# Digital health interventions in strengthening primary healthcare systems in Sub-Saharan Africa: Insights from Ethiopia, Ghana, and Zimbabwe

Tungamirirai Simbini[1]ᵒ*, Emma Adimado[2]ᵒ, Samuel Adjorlolo[3,4]ᵒ, Lorena Guerrero-Torres[5]ᵒ, Prashanth Srinivas[6]ᵒ, Simukai Zizhou[7]ᵒ, Taddese Zerfu[8]ᵒ

**1** Faculty of Medicine and Health Sciences, University of Zimbabwe P.O.Box A178 Avondale, Harare, Zimbabwe, **2** Ghana Health Service, Private Mail Bag, Ministries, Accra, Ghana, **3** Department of Mental Health, School of Nursing and Midwifery, University of Ghana, Accra, Ghana, **4** Research and Grant Institute of Ghana, Legon, Accra, Ghana, **5** Alliance for Health Policy and Systems Research, World Health Organization, Geneva, Switzerland, **6** Center for Health Systems, Institute of Public Health Bengaluru, Bengaluru, India, **7** Ministry of Child Care, Causeway, Harare, Zimbabwe, **8** International Food Policy Institute (IFPRI), Addis Ababa, Ethiopia

ᵒ These authors contributed equally to this work.
* tsimbini@medsch.uz.ac.zw, simbinit@gmail.com

## Abstract

Digital Health Interventions (DHIs) refer to discrete technological functionalities designed to achieve specific objectives in addressing health system challenges. These interventions are considered tools for strengthening health systems, particularly in low- and middle-income countries. This study consolidates findings from Ethiopia, Ghana, and Zimbabwe, examining how three distinct digital health applications with varying intervention components implemented in primary healthcare settings contribute to health system strengthening. The interventions analyzed include Ethiopia's District Health Information System 2 (DHIS2), Ghana's District Health Information Management System (DHIMS) and the Lightwave Health Information Management System (LHIMS), and Zimbabwe's *Impilo* Electronic Health Record (E-HR) system. In Ethiopia, DHIS2 enhanced health system accountability and data quality by streamlining district-level data aggregation, reporting, and performance monitoring. This led to more informed decision-making and improved resource distribution. In Ghana, DHIMSs functions as a public health-level DHI, facilitating national data-driven performance monitoring, while LHIMS operates at the patient level, supporting patient tracking and management, improving patient workflows and resource tracking. However, a lack of interoperability between these two systems has led to data duplication challenges. Zimbabwe's *Impilo* E-HR, a patient-level DHI, has streamlined clinical workflows, improved information sharing, and enhanced decision-making at the point of care. Despite these successes, challenges persist across the three contexts: infrastructure limitations, high staff turnover, and insufficient user

**Data availability statement:** The data has been uploaded on figtree. The DOI is 10.6084/m9.figshare.30646865.

**Funding:** This work received financial support from the WHO Alliance for Health Policy and Systems Research (Alliance) Zimbabwe (Grant Numbers 2023/1325412 and 2024/1440425), Ghana (Grant Numbers 2023/1330956 and 2024/1442222, and Ethiopia (Grant Numbers 2023/1326656 and 2024/1441908). The Alliance is able to conduct its work thanks to the commitment and support from a variety of funders. These include our long-term support from the Swedish International Development Cooperation Agency (SIDA) and the Norwegian Agency for Development Cooperation (NORAD), as well as designated funding for specific projects within our current priorities. For the full list of Alliance donors, please visit https://ahpsr.who.int/about-us/funders. The funders had no role in study design, data collection and analysis, decision to publish, or preparation of the manuscript.

**Competing interests:** The authors have declared that no competing interests exist.

technical capacity. Interoperability issues, particularly in Ghana and Ethiopia, hinder seamless data exchange, while sustainability concerns such as funding gaps and inadequate government support undermine the systems' full potential. The study findings demonstrate that investments in DHIs in primary healthcare may not result in health systems strengthening without addressing baseline conditions for their implementation and sustainability.

## Author summary

Health systems are conceptualised through the World Health Organization's six building blocks. Digital Health Interventions (DHIs) are actively strengthening these building blocks by improving real-time data utilization, streamlining patient management, and enabling evidence-based decision-making at primary health-care levels. This paper synthesizes findings from three independent studies in Ethiopia (DHIS2), Ghana (DHIMS and LHIMS), and Zimbabwe (Impilo E-HR), each playing a unique role in strengthening their respective health systems. We explore how the DHIs have contributed to key health system functions, highlighting successes in data utilization, patient management, and decision-making. At the same time, we report on the challenges faced by implementers, including infrastructure limitations, interoperability issues, and sustainability concerns. Based on our analysis and contextual knowledge, we offer insights and practical recommendations to overcome these challenges, ensuring that DHIs can maximize their impact on service delivery. Without addressing these barriers, DHIs may not enhance health system resilience and efficiency, and ultimately, their impacts on health outcomes may be hindered.

## Introduction

Digital Health Interventions (DHIs) refer to discrete technological functionalities designed to address specific health objectives through digital health applications [1]. Across various digital health applications, DHIs are increasingly recognized as pivotal tools for strengthening health systems in low- and middle-income countries (LMICs). In Sub-Saharan Africa (SSA), where healthcare systems often grapple with limited resources, fragmented services, and logistical challenges, digital technologies offer innovative solutions to enhance service delivery and health outcomes [2]. The adoption of digital health applications in these countries has led to significant improvements in various aspects of health systems. For instance, electronic health records and mobile health applications have facilitated better data management, improved patient tracking, and enhanced communication between healthcare providers [3,4]. These technologies have demonstrated potential for efficient health service delivery, contributing to better patient outcomes and more robust health systems. By integrating DHIs into their health systems, countries aim to address some of the critical

gaps in their healthcare infrastructure, making strides towards more effective and equitable health systems [5]. Despite the promising advancements, the implementation of DHIs in Sub-Saharan Africa is fraught with challenges. Inadequate infrastructure, limited technological literacy, and data privacy concerns often hinder the effectiveness of these initiatives. Additionally, the sustainability of digital health projects is a recurring challenge, with many interventions facing difficulties in maintaining financial and operational support over the long term [6,7]. These barriers can impede the realization of the potential benefits of DHIs, necessitating a closer examination of their practical limitations and strategies for overcoming them.

This paper synthesizes findings from the implementation of selected DHIs in primary health care settings in Ethiopia, Ghana, and Zimbabwe. These countries received country-specific grants from the Alliance for Health Policy and Systems Research (the Alliance) to study these DHIs. This article synthesizes and consolidates common themes emerging across the three studies. The study from Ethiopia evaluated the District Health Information System 2 (DHIS2) with respect to accountability, accessibility, equity, and data utilization in primary health facilities. The Ghana study assessed the effectiveness of two systems, the District Health Information Management System (DHIMS), a repository of national aggregate data on critical health indicators, and the Lightwave Health Information Management System (LHMIS), a local electronic health record system, on how they improved access to and use of health data by primary health managers. The Zimbabwe study examined the contribution of the *Impilo* electronic health record system on health systems strengthening, emphasizing improvements in care quality and health equity at the primary care level. Therefore, this paper aims to explore common opportunities and challenges associated with integrating DHIs in health systems in Ghana, Ethiopia and Zimbabwe. It provides insights into how various DHIs influence health service delivery, data management, and overall system effectiveness, offering guidance for future implementations and improvements for DHIs in similar primary healthcare settings.

### DHIs implementation context

**Ethiopia.** Ethiopia has made strides in incorporating DHIs into its healthcare system, with a particular focus on reaching rural and underserved areas. The country's diverse geographic and socio-economic environments have provided essential insights into how DHIs can be effectively adapted to different contexts, ranging from urban centers to remote villages. Since 2017, Ethiopia has implemented DHIS 2 as its standard platform for data collection and reporting, which is now successfully operating in over 4,000 healthcare facilities across the nation.

**Ghana.** By 2017, Ghana had implemented two digital health information systems, namely the District Health Information System 2 (DHIS2), locally named the DHIMS, which serves as a repository for aggregate data for the largest implementing agency within the health sector, the Ghana Health Service (GHS) and an electronic health management system known as LHIMS hosted by the Ministry of Health. The DHMIS is deployed in all primary and secondary healthcare facilities within the GHS, while the LHIMS is deployed in the country's district, regional hospitals, and teaching hospitals. These two systems are currently not interoperable. The target users of both systems are all healthcare professionals and policy makers within the health sector. Currently, most public health facilities within the 16 regional capitals are using the LHIMS as a point of care application, while efforts are ongoing to extend to other points of care, like health centers.

**Zimbabwe.** Zimbabwe adopted DHIs for patient tracking in 2013, initially focusing on monitoring patients on antiretroviral therapy (ART) [8]. In 2016, the country piloted *Impilo*, a locally developed web-based electronic health record (E-HR) system, to expand patient tracking beyond HIV/AIDS treatment. By 2023, *Impilo* had become the primary E-HR system in public health facilities, implemented in 1,055 of the 1,800 primary and secondary care facilities nationwide. The system includes 18 modules covering various clinical care departments, such as Patient Registration, General Outpatients Consultation, HIV/AIDS/TB/STIs, Pharmacy and Commodity Tracking [9], effectively supporting all health system building blocks at these levels of care. The system is built on the Open Health Information Exchange (Open HIE) framework

and integrates Health Level 7 Fast Interoperability Resources (HL7 FHIR), enabling seamless interoperability with other systems, including laboratory information management systems and automated reporting in DHIS2 [9].

## Research methods

### Ethical approvals

**Ethiopia:** Ethical approval was obtained from five regional Institutional Review Boards (IRBs) under the respective health bureaus: Addis Ababa City Administration and the four regions (Oromia, Amhara, Sidama, and Somali). **Ghana:** The study received ethical approval from the Ghana Health Service Ethical Review Committee (GHS-ERC003/04/23). **Zimbabwe:** The study was approved by the Joint Research and Ethics Committee JREC at the University of Zimbabwe, the Ministry of Health through the Permanent Secretary and the Medical Research Council of Zimbabwe under study ID MRCZ/A/3127.

### Ethics statement

All participants in Ethiopia, Ghana, and Zimbabwe provided informed consent before taking part in the study. The study's purpose, the voluntary nature of participation, confidentiality safeguards, and the right to withdraw at any time were explained to each individual. Both verbal and written consent were obtained and recorded at the start of each interview, consistent with ethical guidelines for minimal-risk research. Each of the three country studies employed a mixed-methods design and was followed by an independent analysis of that country's findings.

### Methodology

The Ethiopian study employed a mixed-methods approach to evaluate the impact of DHIS2 on maternal and child health (MCH) outcomes in Ethiopia. The research was conducted across four regions (Oromia, Amhara, Sidama, and Somali) and an administrative city (Addis Ababa) - selected to represent the country's socio-economic and geographic diversity. The primary objective was to examine how DHIS2 strengthened primary healthcare systems through data availability, accessibility, and evidence-based decision-making for MCH.

Quantitative data were collected from 132 health facilities, involving health workers, facility managers, and regional health officials from Primary Health Care Units (PHCU). This included a review of retrospective performance reports and electronic health data from 2013 to 2022, using 2017 as a point of divergence to assess trends. These data were complemented by qualitative insights from 30 Key Informant Interviews (KIIs) and 16 Focus Group Discussions (FGDs), which explored user perspectives on DHIS2's usability and effectiveness.

Quantitative data were analyzed using an Interrupted Time Series Analysis (ITSA) to model changes in key MCH indicators following DHIS2 implementation. For the qualitative data, transcripts from KIIs and FGDs underwent thematic analysis, aided by ATLAS.ti software. This tool facilitated the systematic organization, coding, and interpretation of data to identify recurring themes concerning accessibility, data quality, and equity.

**Ghana** conducted a sequential mixed-methods study to assess how DHIMS and LHIMS enhanced access to and use of healthcare data by primary health care (PHC) managers. The study involved 125 PHC managers from 10 district hospitals and one regional hospital in the Greater Accra Region. The team used purposive sampling, selecting all secondary healthcare facilities located in the Greater Accra region that use both the DHIMS and LHIMS. The study employed purposive sampling to select 125 PHC managers from participating facilities who possessed relevant experience with DHIMS and LHIMS, and were willing to provide information on these DHIs. The first phase involved the administration of survey questionnaires using cross-sectional design to generate quantitative summaries of the application of DHIMS and LHIMS from hospital unit managers/in-charges. This was followed by a qualitative exploratory approach involving individual in-depth interviews (IDIs) to obtain in-depth insights into the DHIMS and LHIMS from PHC managers. Quantitative data were analyzed using descriptive statistics in STATA. For the qualitative component, transcripts from KIIs and IDIs were manually coded using an inductive thematic analysis approach, drawing on Braun and Clarke's six-step model [10]. This

iterative process involved open coding, constant comparison of emerging categories, and development of higher-order themes to elucidate contextual factors surrounding DHIMS and LHIMS implementation. Findings from the quantitative and qualitative strands were triangulated to provide a more comprehensive and nuanced understanding of how DHIMS/LHIMS is applied in practice. The research focused on the contributions of DHIMS and LHIMS to data for decision-making, examined interoperability issues between the systems, and identified challenges associated with the healthcare data provided by these platforms.

**Zimbabwe** conducted a quasi-experimental, non-randomized mixed-methods study to assess the impact of E-HR on health systems building blocks, quality of care, and implementation challenges in resource-limited settings. The study was conducted in two purposively sampled provinces. In each province, an E-HR implementing district and a non-implementing district were selected. The selected implementing districts began using E-HR in 2021. Thirty-two health facilities were sampled, twenty (20) from E-HR implementing sites and twelve (12) from non-implementing sites. District hospitals were automatically included. The study population were 21 key informants (district health executive), 37 health care providers (nurses, doctors, pharmacists and laboratory scientists) and 173 patients receiving care. Participation was voluntary, and individuals were enrolled after providing informed consent. The study investigated E-HR contributions on each of the health system building blocks, quality of care, and the role of E-HR in improving decision-making, equity, quality, and efficiency, while documenting lessons from large-scale E-HR deployment. Qualitative data were collected through audio recordings and manually transcribed into Microsoft Excel sheets, organized by question and response. Each question was reviewed and summarised to identify common themes, which were further refined using a large language model for thematic synthesis and summary generation. Quantitative descriptive statistics, primarily frequencies, were computed in Excel.

In this study, we critically examined the findings across these three studies to identify shared patterns and identify learnings across these sites in order to contribute towards a better understanding of the DHI implementation in PHC in other similar settings.

## Results

We categorized the findings from the three countries based on the applicable health systems building blocks.

### Health information systems

Findings from the three studies indicate a widespread adoption of digital health tools, particularly DHIS2, across primary and secondary healthcare facilities. In all country settings, DHIs serve as the core reporting platform for aggregate health data.

The Ethiopian study detailed the critical role of DHIS2 in key functions: among reporting facilities, 99% use it for data reporting, 96% for program monitoring, 87% for resource allocation, and 67% for target setting. This functionality has enhanced healthcare management by streamlining reporting processes and enabling effective resource distribution to the areas of greatest need. Consequently, DHIS2 has become instrumental in facilitating evidence-based decisions that improve service delivery.

Patient-level information systems are seeing an increased uptake in both Ghana and Zimbabwe. In Ghana, all primary and secondary facilities have adopted the LHIMS. In Zimbabwe, 56% of primary and secondary facilities have adopted E-HR. Data is cascaded from facility to national level in all cases. In Ghana, data cascading was facilitated by the DHIMS. There is increased reliance on these DHIs by service providers, and decision makers alike. In Ethiopia, DHIS2 has strengthened the health system by enhancing data quality, availability, and accessibility. It streamlines collection and enables real-time reporting, ensuring consistent and accurate data while reducing errors. This improved access empowers policymakers to make evidence-based decisions and allocate resources more effectively. Furthermore, the system promotes accountability by tracking performance and ensuring public health strategies are data-driven. Similarly, in Ghana,

both DHIMS and LHIMS have been integral to data-driven decision-making for PHC managers, particularly in forecasting, monitoring, and evaluating performance. In Zimbabwe, the E-HR has improved information sharing among health practitioners, enhanced data integrity through audit trails, and strengthened decision support at the point of care, as noted by the following responses:

> "*Increased credibility in data, what you enter into the E-HR is what is displayed in the E-HR unlike when using paper*" [Zim Key Informant]

Accessing patient level information was also noted to improve

> "*Reduces time taken to find the patient's record*" [Zim Nurse Manager]

It has also streamlined national reporting requirements, reducing the number of manual aggregate reports and registers generated, hence leaving more time available to patient care:

> "*It can improve workforce effectiveness…, no more registers required to be written which consumes most of the time.*" [Zim Key Informant]

> "*It is easy now to obtain summary data to know how your department is running. It is better than the paper system*"[Gha Key Informant]

Integration of E-HR with DHIS2 has reduced the reporting requirements on nurses and increased the reliability and timeliness of reported data.

## Service delivery

Ethiopia and Zimbabwe both reported improvements in service delivery due to their respective DHIs. In Zimbabwe and Ghana, the E-HR system and LHIMS have streamlined clinical workflows, particularly in prescriptions, reducing patient wait times and increasing the efficiency of service delivery.

> "*E-HR 'Impilo' has streamlined our processes and improved overall service delivery.*" [Zim Key Informant]

> "*Because of the LHIMS, we do not need to give folders to patients to go to the lab or pharmacy to obtain their drugs. The order is issued at the ward and is received immediately by these units. Work is streamlined*"[Gha Key Informant]

The availability of real-time patient records has enhanced patient experiences by minimizing health care worker time spent on documentation tasks. However, not all users share the same sentiments; some users noted that using the E-HR system became more time-consuming during periods of high patient volumes. The slow response times of these applications, during high patient volume reduces the effectiveness of E-HR at point of care:

> "*The way I see it, it's a burden. In the case of a patient who comes for HIV testing, it takes 5–10 minutes to read the records and yet the patients are waiting in the queue, we end up resorting to shortcuts because I still have to attend to others….* [Zim Health Worker].

In Ethiopia, DHIS2 has similarly contributed to more efficient healthcare delivery by improving data accuracy and timeliness, which are crucial for effective resource allocation and service planning. For instance, a PHCU head from one region highlighted the significant improvement in timeliness after DHIS2 implementation, saying:

*"Before DHIS2, we struggled with tracking data timeliness. After implementation, the transformation was clear - our data management reports are now more timely than ever."* [Eth PHCU head]

## Health workforce

All three countries have reported the increased use and integration of systems within the various service departments, including clinical care workflows. The implementation of DHIS2 in Ethiopia and DHIMS in Ghana has significantly improved accountability within the healthcare sector and health workers. These systems enable policymakers at district/regional and national levels to monitor resource utilization and assess the impact of healthcare interventions more effectively.

However, challenges related to technical capacity have emerged as a significant barrier to the full utilization of DHIs. A shortage of health personnel adequately trained in digital health in general, and in digital applications specifically, was noted in all three countries, hindering optimal system operation. Even among trained staff, the lack of continuous technical support of the digital application exacerbates these challenges. Furthermore, Zimbabwe and Ethiopia reported high staff turnover for those trained in the use of the DHIs.

Ghana noted 88% of interviewed PHC managers relied on LHIMS data, while 70% of them obtained summary data from DHIMS. About 73% reported that the DHIMS/LHIMS was easy/very easy to access and majority of the participants, 86% indicated that the DHIMS was presented in an easy-to-understand format, compared with the LHIMS (63%). More importantly, about 96% (n = 88) and 81% (n = 95) noted the information in DHIMS and LHIMS to be *Very Useful* and *Useful*, respectively.

*"It helps us to make decisions. For example, if I know the number of deliveries that I have had for a period it will help me to improve my system. When I say system, I mean both managerial as well as logistical"* (Gha-PHC manager)

Ghana's experience highlighted the struggle of older healthcare professionals, particularly in managerial roles, with accessing and utilizing data from LHIMS, despite its availability. Only 13% of the respondents self rated their computing competence as Excellent and 50% rated themselves as average or good.

In Zimbabwe, E-HR was found to significantly enhance decision support for health workers, with features like the e-Partograph automating patient tracking during labor and delivery.

*"Use of e-Partograph. It has improved so much, the nurse does not have to forget to listen to the foetal heart because it gives alerts and it flicks when the time comes to say now you are supposed to go listen to the foetal heart, to do the vaginal examination, blood pressure."*[Zim Health Worker]

The integration of clinical practice guidelines within the E-HR system standardized the quality of care and fostered better teamwork through shared patient management histories.

*"Improved on communication since it allows colleagues to know how I have managed the patient and what further services are needed"* [Zim Health Worker]

However, a notable challenge was the lack of doctor participation in entering patient records, which diminished the overall effectiveness of the system.

*"I cannot say it has improved. When we capture the observations or vitals when we refer the patient to the doctor, most of the time, they don't input the data in the system, so there is a missing link."* [Zim Health Worker]

## Medicines and supplies

Zimbabwe's E-HR system, through its Pharmacy and Commodity Tracking module, provided real-time tracking of medicines, including stock levels, usage, and expiration dates, and monitored minimum and maximum stock levels.

> "*It shows a breakdown of medicines that are in short supply. Rarely do you miss to order medicines. It tells us what we have, for example we have 7 sachets of paracetamol in stock but only 2 have not expired*" [Zim Nurse Manager]

While this improved inventory management, stock-outs persisted due to broader supply chain challenges that were beyond the scope of the E-HR system.

> "*We order from National Pharmacy (Nat Pharm), but the deliveries come incomplete or with shortages or no deliveries at all*" [Zim Nurse Manager]

Similar issues were noted in Ethiopia, where DHIS2 improved resource allocation but was hampered by infrastructure and technical capacity limitations characterized by high staff turnover, and poor stakeholder coordination.

## Health system financing

Findings in the domain of health financing were less explored in all three countries. However, all three countries reported findings on the use of DHIs in resource tracking and allocation, an indirect contribution towards efficient utilisation of resources and hence cost savings. Zimbabwe and Ethiopia noted that despite the potential of DHIs in resource tracking and allocation, limited funding in the health system will continue to result in resource shortages such as continued drug stock outs (Zimbabwe), and high staff turnover (Zimbabwe and Ethiopia).

## Governance and leadership

The findings from the three studies note the increased accountability among decision makers at each administrative level, hence strengthening health systems governance structures. In Ethiopia there is 96% usage of DHIS2 for program monitoring, and 87% for resource allocation at primary health units, increasing the availability of programme data strengthening decision making and accountability at an administrative level. In Ghana, the data suggested an increase in accountability using the DHIMS system, with 70% of PHC managers obtaining summary data from DHIMS. In Zimbabwe, the availability of data in the medicines supply chain to nurses in the frontline has demonstrated the increased accountability of health services.

## Cross-cutting issues

All three countries faced significant challenges in DHI implementation, particularly related to infrastructure. In Ethiopia, Ghana and Zimbabwe, frequent power outages, unreliable internet connectivity, and the theft of solar panels (Zimbabwe) and tablets (Zimbabwe) disrupted the functionality of the DHIs.

> "*Here the challenges are no electricity, solar panels stolen, Wi-Fi never worked here because there is no power. E-HR has its own problems which are not common in routine practices*" [Zim Nurse Manager]

The use of personal devices to access LHIMS poses a major security issue in Ghana. Likewise, multiple individuals can have unrestricted access to these personal devices, raising questions about data security and confidentiality.
Key informants highlighted the need for a more comprehensive systems infrastructure to adequately support the implementation of E-HR in Zimbabwe.

*"The existing infrastructure does not support the new system well."*[Zim Key Informant]

Furthermore, the study in Ethiopia found that infrastructure in many rural and remote areas remains inadequate, with unreliable internet connections and frequent power outages disrupting system functionality. These challenges hindered the utilization of DHIS2, preventing healthcare providers from accessing or updating information in real-time, as healthcare workers were often unable to enter data or generate reports promptly, leading to delays in decision-making.

In Ghana, older healthcare professionals, often in managerial positions, struggled with retrieving information from LHIMS, despite it being readily available for decision-making. The ease of accessing information was identified as a contributing factor to delays in data utilization by PHCs in Ghana. For Zimbabwe integration of E-HR in clinical workflows particularly within high-volume sites, was a challenge. Ethiopia noted DHIS2 challenges related to data quality and interoperability. In some cases, the data entered into DHIS2 was incomplete, inaccurate, or inconsistent, undermining its reliability for decision-making. There are also concerns about how well DHIS2 interacts with other health information systems, both within Ethiopia and at the regional level. Interoperability problems make it difficult to consolidate data from multiple sources, hindering the integration of health information across various platforms and preventing a comprehensive view of the healthcare system. For Ghana, the dual use of DHIMS and LHIMS without interoperability between these two systems has resulted in duplication and data inaccuracies. In Zimbabwe, although E-HR is interoperable with DHIS2, the reliance on paper record systems as a backup during E-HR downtime has led to inconsistent and inaccurate data. This is primarily due to the high human resource costs associated with updating historical data into the electronic system once it is operational. The sustainability of DHIS2 was a significant concern in Ethiopia. The system's long-term success depended on adequate financial resources, continued government commitment, and ongoing support from development partners.

## Discussion

Our paper examined how established DHIs in three countries have contributed to one or more of the health systems building blocks. In summary across Ethiopia, Ghana, and Zimbabwe, DHIs are becoming central to how primary health care data are collected and used, improving data quality, timeliness, and accountability for decision-making. These systems have institutionalised data use and performance tracking. However, there are persistent challenges: unreliable infrastructure, staff turnover, and limited digital skills, which reflect deeper system constraints that limit their full potential.

Most of the existing literature focuses on one or more building blocks, with the majority of studies emphasizing health service delivery. In a scoping review of DHIs and health systems strengthening in sub-Saharan Africa [6], health service components were overwhelmingly represented, accounting for 82% (n = 603 out of 738 DHIs included), while leadership and governance were the least represented, comprising only 0.4% (n = 3) of studies. Notably, 34% (n = 252) of interventions were found to strengthen more than one of the six building blocks. Among all DHIs, the largest proportion (84%) were primarily focused on data collection and analysis, rather than directly improving service provision. In contrast to most previous studies, this paper reports on DHIs across multiple health system building blocks in all three countries, highlighting their contributions not only to data reporting and service delivery but also to medical products and supply, health workforce, and, by inference, to financing as well as governance and leadership at both primary care and national levels.

According to the WHO Digital Health Atlas, DHIS2 is the most widely adopted DHI in Africa, with 56% of all DHIS2 implementations on the continent [11]. All three countries of this study use DHIS2 for health data reporting, with implementation spanning from primary care facilities to the national level, providing performance indicator data at each stage of care. Almost all national reporting in Ethiopia (99%) and Ghana (70%) is conducted through DHIS2. Of the three study countries, only in Zimbabwe is DHIS2 integrated with the electronic health record system, enhancing efficiencies in health data management and enabling real-time reporting. Therefore, DHIS2 is now institutionalised as the main reporting tool for aggregate reporting requirements at primary, secondary and national levels. Patient level systems are also taking a significant role in data management at points of care, with Zimbabwe having a penetration of 56% of all its public sector

primary care facilities and Ghana reporting 88% PHC managers rely on patient level information systems in the sampled facilities. In Zimbabwe, it is the nurse managers and nurse caregivers at primary level using E-HR in their daily practices. The motivation for uptake of these DHIs includes improved health data availability, enhanced credibility of data, and the facility to monitor programmes, resources and performance in near real-time. These motivators have been documented in other settings, including Kenya [12], Uganda [13] and South Africa [14]. The wide spread access to and the reduced turnaround time for accessing health information has been associated with improved resource allocation and real-time decision-making at administrative and policy levels. At the point of care, faster access to patient information has also shortened clinical decision cycles, enabling timely interventions and enhancing service delivery across the health system continuum. Collectively, these improvements contribute to a strengthened continuum of care and more efficient coordination between different service delivery points. A separate study by Mensah et al (2024) [15] documented similar improvements across the continuum of care.

Although DHIs have improved service delivery at the point of care, their benefits are not equitably realised across all cadres of health workers. In Zimbabwe, the study reported limited clinician engagement, notably low doctor participation in updating electronic patient records; this creates gaps in the completeness of digital patient data. These human-factor barriers, clinician non-participation and limited digital literacy, are major bottlenecks that constrain the full potential of DHI implementation. A systematic review by Odekunle et al. (2017) [16] on barriers to electronic health record adoption in sub-Saharan Africa highlighted digital literacy gaps among health workers as a critical limiting factor, alongside infrastructural challenges and resistance to change. This is consistent with evidence from Ghana, where studies have shown that limited computer proficiency, especially among older primary healthcare managers, constrains the use of patient-level electronic systems and undermines the completeness and quality of health data [15,17]. These findings show that digital health transformation efforts require targeted digital literacy interventions for healthcare workers, particularly older, less digitally savvy individuals.

Odekunle et al. (2017) [16] noted infrastructure challenges such as electricity and internet availability are a limitation to the adoption of electronic health records. Achampong (2012) emphasized that improvements in infrastructure, including computer hardware, networking capacity, and reliable telecommunication are essential for the resilient implementation of E-HR systems. In Zimbabwe, these findings are echoed by users who highlight that E-HR functions as an independent system that can exacerbate pre-existing health system challenges if it is not well supported. The unstable electricity and internet connectivity mean that disruptions in either service can halt patient care or force a reversion to paper-based documentation. Once paper records are used, subsequent transcription into the electronic system becomes an expensive and time-consuming task, introducing significant human resource burdens and hence impacting on the longitudinal patient record. Hence, our study highlights the importance of implementing a holistic DHI supportive ecosystem for a resilient digital implementation, and potential impacts of not doing so.

The lack of interoperability as reported in the Ghana study, results in fragmentation of information across different service points (LHIMS) and policy makers (DHIMS) as well as the lack of cohesion between caregivers (nurses and doctors) as reported in the Zimbabwe study, makes it challenging to consolidate the information available for a holistic response at the point of care and decision making at policy level, a finding that has been noted by Atasoy H et al [18].

DHI solutions for patient care must have a shorter turnaround time in data retrieval to cater for high patient volumes. The Zimbabwe study reported high patient volumes and associated workload pressures inhibiting optimal service delivery, compounded by slow response times and system delays at the point of care would discourage caregivers from fully utilizing E-HR systems during patient encounters. To address the long latency of E-HRs at point of care, E-HR implementers should focus on decentralized solutions where data is locally available and synchronized during off-peak hours. Some strategies to mitigate E-HR systems resilience include deploying localized servers at healthcare facilities to securely store and process patient data, particularly for frequently accessed cases. Complementary non-technical strategies include implementing structured patient scheduling and effective management of non-emergency cases by hospital

administrators. These measures can improve service efficiency and mitigate the impact of limited human and material resources in high-volume healthcare facilities [19].

Gaps in digital skills, such as those reported in Ghana where older primary-care managers and less digitally literate staff struggle to use patient-level systems, have reduced system uptake and data quality [20]. Strengthening digital competencies among health workers, especially older cadres, is essential for inclusivity and effective DHI use. Targeted capacity building through structured training, mentorship, and refresher programs can support this process [21,22]. Integrating DHI training into health education curricula will prepare future professionals to use these tools from the start of their careers [23,24]. Institutionalizing DHIs within standard operating procedures and defining clear roles for healthcare cadres will embed digital tools in routine care [25,26]. Without strategies to retain trained personnel, digital health investments risk being lost, limiting the impact of DHIs on primary health-care delivery [27].

Regarding the medicines and supply building block, though DHIs have been recognized for their positive impact on resource tracking [28,29], DHIs alone are insufficient. For their full potential to be realized, they must be supported by broader efforts to strengthen other health system building blocks. The persistence of stockouts, driven by the underfunding of essential commodities, underscores the need for better resource allocation and overall system strengthening. To maximize the impact of DHIs, policymakers should strengthen all health system building blocks, ensuring reliable commodity availability, efficient supply chains, skilled human resources, and sustainable financing.

Regarding health financing as a building block, these three studies did not examine this area directly. However, the improvements in resource utilization have indirectly benefited healthcare financing by enhancing efficiency and reducing costs associated with resource wastage and system leakages. Therefore, further research is needed to better understand and possibly quantify how DHIs contribute to the health financing building block and their overall impact on cost-effectiveness in healthcare systems.

These findings present some opportunities for policy makers, funders and implementers.

**Policymakers** should prioritize strengthening all health system building blocks while integrating DHIs as part of a holistic approach to health systems strengthening. DHIs ought to be regarded as enablers that enhance existing systems rather than as standalone or parallel implementations. In addition, there should be deliberate advocacy to incorporate digital literacy training into standard medical training curricula to ensure that all health workers possess the necessary competencies to effectively use digital tools. The World Health Organization (WHO) emphasizes that cultivating a digitally literate health workforce is essential for improving efficiency and overall system performance [30]. Institutionalizing DHIs within health service delivery, particularly at the primary health care level, remains a critical step that requires continued support and sustainability. However, limited and inconsistent funding for DHIs poses a potential threat to the continuity of health systems strengthening efforts. To ensure long-term impact, these interventions should be embedded and financed within broader national health and digital transformation strategies.

**Funders** should support DHIs as comprehensive programs that emphasize systems integration and alignment within the broader health system. Investments should focus on building sustainable ecosystems of interoperable digital solutions, supported by strong human resource capacity and long-term operational frameworks. Sustained funding is essential for the institutionalization of DHIs, ensuring that they are embedded within the overall health system rather than implemented as short-term, standalone projects.

**Implementers** should ensure interoperability across platforms, address infrastructure limitations to guarantee the availability of mission-critical DHIs, and optimize user interfaces and user experiences for seamless data entry at the point of care. Additionally, robust change management strategies should be embedded to support user adoption and system sustainability.

## Conclusion

Our study, examining DHIs from three African countries, showed the potential of DHIs in strengthening health systems in resource-limited settings. The cases of Ethiopia, Ghana, and Zimbabwe demonstrate that DHIs, such as DHIS2, LHIMS, and the *Impilo* EHR system, enhance data management, accountability, and decision-making. These systems contribute to improved service delivery, and resource allocation., However, critical challenges, including infrastructure deficiencies, technical capacity gaps, high staff turnover, and interoperability limitations, continue to hinder the full realization of their benefits.

## Limitations

The purpose of combining these three independent studies was to identify common themes and cross-cutting opportunities across different country contexts. However, several limitations should be noted. Each study was conducted by separate research teams, using distinct methodologies, analytical frameworks, and timelines. As a result, direct comparisons between the studies may be constrained by variations in study design, data collection tools, and contextual factors. Furthermore, while efforts were made to synthesize the findings objectively, the integration of independently generated data introduces a degree of interpretive bias. Despite these limitations, the synthesis provides valuable insights into shared experiences and lessons for digital health implementation in the three African country contexts.

## Author contributions

**Formal analysis:** Tungamirirai Simbini.

**Writing – original draft:** Tungamirirai Simbini, Emma Adimado, Samuel Adjorlolo, Simukai Zizhou, Taddese Zerfu.

**Writing – review & editing:** Tungamirirai Simbini, Lorena Guerrero-Torres, Prashanth Srinivas, Taddese Zerfu.

## Acknowledgments

We would like to acknowledge the Alliance for Health Policy and Systems Research (Alliance), and the respective Ministries of Health in the respective countries and all the participants who took part in these studies. The authors alone are responsible for the views expressed in this article and they do not necessarily represent the views, decisions or policies of the institutions with which they are affiliated.

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
