## [Decision Letter · Decision Letter 0]

29 Jul 2025

Response to Reviewers
Revised Manuscript with Track Changes
Manuscript
**Journal Requirements:**

i.. State the initials, alongside each funding source, of each author to receive each grant.

ii. State what role the funders took in the study. If the funders had no role in your study, please state: “The funders had no role in study design, data collection and analysis, decision to publish, or preparation of the manuscript.”

2. Please ensure that your Ethics Statement is available in its entirety at the beginning of your Methods section, under a subheading 'Ethics Statement'.

3. We ask that a manuscript source file is provided at Revision. Please upload your manuscript file as a .doc, .docx, .rtf or .tex.

4. In the online submission form, you indicated that “The data is largely qualitative interviews in Excel Sheets. These can be made available online as requested”. 

3. Uploaded as supplementary information.

**Additional Editor Comments (if provided):**
**Reviewers' Comments:**

**Comments to the Author**

1. Does this manuscript meet PLOS Digital Health’s publication criteria?

Reviewer #1: Yes

Reviewer #2: No

Reviewer #3: Yes

2. Has the statistical analysis been performed appropriately and rigorously?

Reviewer #1: Yes

Reviewer #2: N/A

Reviewer #3: I don't know

3. Have the authors made all data underlying the findings in their manuscript fully available (please refer to the Data Availability Statement at the start of the manuscript PDF file)?

Reviewer #1: Yes

Reviewer #2: No

Reviewer #3: No

4. Is the manuscript presented in an intelligible fashion and written in standard English?

Reviewer #1: Yes

Reviewer #2: Yes

Reviewer #3: Yes

Reviewer #1: The manuscript is essential in its provision of a relevant analysis of Digital Health Interventions in three African countries. It offers valuable insights into their role in strengthening health systems. The study has an excellent structure with good methodology while addressing critical gaps in literature. Please find a few comments below for your consideration.

Sampling Details: Provide more information on the sampling strategies (e.g., criteria for selecting regions/facilities in each country) to enhance reproducibility.

Data Analysis: Clarify how qualitative data (e.g., interviews, FGDs) were analyzed (e.g., software used, coding framework). For quantitative data, consider including statistical measures (e.g., p-values, confidence intervals) where applicable.

Policy Recommendations: Strengthen the conclusion by explicitly linking findings to actionable recommendations for policymakers, funders, and implementers.

In addition, kindly re-read the manuscript. Some of the sentences are a bit too long and need improvement in conciseness.

Reviewer #2: There are several ambiguities across the manuscript which needs to be solved before sending for peer reviewing. The structure and design of the article is unclear. It is not clear why three separate studies from three countries were combined and presented in the form of one manuscript. Normally, if the reader is interested in the topic, he/she will refer to the three original articles. Why were the three countries chosen and based on what criteria was this choice made? Detailed comments have been included in the the attached file.

Reviewer #3: - Underlying data isn't available yet, although the authors indicated that it can be made available as requested.

- The methods section is fair enough, however, it would help to include more details of specific tools/software used for analyses. Also, for the Ethiopia study, the exact number of key informant interview and focus group discussion participants across the 5 regions were not indicated.

To the authors: Highly relevant study, however it could benefit from more evidence support. For instance, in the strategies recommended in the Discussion section, it would help to include references to studies that show how these strategies have worked in similar contexts.

**Do you want your identity to be public for this peer review?** For information about this choice, including consent withdrawal, please see our Privacy Policy

Reviewer #1: No

Reviewer #2: **Yes: ** Hadi Ghasemi

Reviewer #3: No

**Figure resubmission:****Reproducibility:** To enhance the reproducibility of your results, we recommend that authors of applicable studies deposit laboratory protocols in protocols.io, where a protocol can be assigned its own identifier (DOI) such that it can be cited independently in the future. Additionally, PLOS ONE offers an option to publish peer-reviewed clinical study protocols. Read more information on sharing protocols at https://plos.org/protocols?utm_medium=editorial-email&utm_source=authorletters&utm_campaign=protocols

---

## [Editor Report · Decision Letter 1]

25 Nov 2025

Digital Health Interventions in Strengthening Primary Healthcare Systems  in Sub-Saharan Africa: Insights from Ethiopia, Ghana, and Zimbabwe

PDIG-D-25-00230R1

Dear Dr Simbini,

We are pleased to inform you that your manuscript 'Digital Health Interventions in Strengthening Primary Healthcare Systems  in Sub-Saharan Africa: Insights from Ethiopia, Ghana, and Zimbabwe' has been provisionally accepted for publication in PLOS Digital Health.

Best regards,

J. Mark Ansermino, MBBCh

Section Editor

PLOS Digital Health

**Additional Editor Comments (if provided):**

I would suggest the authors proof read the manuscript one more time.